# Spectroscopy of bulk and few-layer superconducting NbSe$_2$ with van der Waals tunnel junctions

T. Dvir [1], F. Massee[2], L. Attias[1], M. Khodas[1], M. Aprili[2], C.H.L. Quay[2] & H. Steinberg [1]

Tunnel junctions, an established platform for high resolution spectroscopy of superconductors, require defect-free insulating barriers; however, oxides, the most common barrier, can only grow on a limited selection of materials. We show that van der Waals tunnel barriers, fabricated by exfoliation and transfer of layered semiconductors, sustain stable currents with strong suppression of sub-gap tunneling. This allows us to measure the spectra of bulk (20 nm) and ultrathin (3- and 4-layer) NbSe$_2$ devices at 70 mK. These exhibit two distinct superconducting gaps, the larger of which decreases monotonically with thickness and critical temperature. The spectra are analyzed using a two-band model incorporating depairing. In the bulk, the smaller gap exhibits strong depairing in in-plane magnetic fields, consistent with high out-of-plane Fermi velocity. In the few-layer devices, the large gap exhibits negligible depairing, consistent with out-of-plane spin locking due to Ising spin–orbit coupling. In the 3-layer device, the large gap persists beyond the Pauli limit.

[1] The Racah Institute of Physics, The Hebrew University of Jerusalem, Jerusalem 91904, Israel. [2] Laboratoire de Physique des Solides (CNRS UMR 8502), Bâtiment 510, Université Paris-Sud/Université Paris-Saclay, 91405 Orsay France. Correspondence and requests for materials should be addressed to H.S. (email: hadar@phys.huji.ac.il)

Superconductors of the transition metal dichalcogenide (TMD) family have seen a revival of interest subsequent to developments in device fabrication by mechanical exfoliation[1–7]. The isolation of ultrathin NbSe$_2$[1,2] has yielded indications of a Berezinskii–Kosterlitz–Thouless transition, which occurs in 2D superconductors. NbSe$_2$[1], gated MoS$_2$[5,6], and gated WS$_2$[7] devices also remain superconducting in in-plane magnetic fields well beyond the Pauli limit $H_P = \Delta / (\sqrt{g} \mu_B)$[8,9]. (Here, $\Delta$ is the superconducting energy gap, $g$ is the Landé g-factor and $\mu_B$ the Bohr magneton.) This is likely due to Ising spin-orbit coupling (ISOC): The broken inversion symmetry of the monolayer TMD in the plane is expected to lead to the formation of Cooper pairs whose constituent spins are locked in the out-of-plane direction, in a singlet configuration. Interestingly, zero-resistance states have been observed in parallel magnetic fields exceeding the Pauli limit even in few-layer devices, where inversion symmetry is recovered[1]. This suggests that the inter-layer coupling is not strong enough to overcome the out-of-plane spin-locking due to ISOC, perhaps in part due to the presence of spin-layer locking[10].

These previous studies[1–7] used in-plane electronic transport at high magnetic field and temperatures close to the critical temperature, $T_c(H = 0)$, to determine the upper critical field $H_{c2\parallel}$, which depends on the magnitude of the spin-orbit field $H_{SO}$. Tunneling spectroscopy can provide complementary information. For example, tunneling and other probes (see ref.[11] and references therein) suggest that bulk NbSe$_2$ is a 2-band superconductor. The role of both bands in ultra-thin devices, and their response to magnetic field, can be addressed by tunnel spectroscopy but not by electronic transport (which addresses the condensate) or heat conductivity measurements[12]. The latter, specifically, would be shorted by the substrate. Tunneling can also probe the effect of magnetic field at the full temperature range and in fields ranging from zero to $H_{c2\parallel}$.

To carry out tunneling measurements on devices of variable thickness, at variable magnetic field conditions, it is necessary to develop a device-based architecture suitable for integration with TMDs. Oxide-based tunnel barriers, such as those used since the days of Giaever[13], have now reached technological maturity; however, there is a limited number of oxides which form high-quality insulating, non-magnetic barriers, and they do not grow well on all surfaces. It is therefore of interest to explore alternatives based on van der Waals (vdW) materials[14], ultrathin layers of which can be precisely positioned on many surfaces[15]. Indeed, such barriers have proven effective when integrated with graphene[16–18] and appear to be promising candidates for integration with TMD superconductors[19].

In this work we study tunneling devices utilizing ultrathin TMDs as barriers. We find that these devices exhibit a hard gap with strong sub-gap conductance suppression. In the spectra of bulk NbSe$_2$, we clearly resolve the presence of two superconducting order parameters, which exhibit a distinct evolution upon application of in-plane magnetic fields. Spectra of ultrathin NbSe$_2$ are stable against the application of magnetic fields, even beyond the Pauli limit[8,9].

## Results

**Hard gap van-der-Waals tunnel junctions**. We fabricate Normal-Insulator-Superconductor (NIS) tunnel junctions with either MoS$_2$ or WSe$_2$—both van der Waals (vdW) materials—as the insulating barrier. The barrier material is placed on top of 2H-NbSe$_2$ (hereafter NbSe$_2$), a vdW superconductor with bulk $T_c \approx$ 7.2 K, using the 'dry transfer' fabrication technique[14,15]. Crucially, such heterostructures can comprise NbSe$_2$ flakes of varying thicknesses, from bulk ($\geq 6$ layers) to few-layer, often within a single device. Figure 1a shows a typical junction consisting of a 20 nm-thick NbSe$_2$ flake partially covered by a 4–5 layer thick MoS$_2$ barrier (cf. Supplementary Note 1 and Supplementary Fig. 1). The junction has an area $A = 1.6\ \mu m^2$, and we evaluate its transparency to be $\mathcal{T} \sim 10^{-8}$ (Supplementary Note 2 and Supplementary Fig. 2). Figure 1b shows the differential conductance $G = dI/dV$ as a function of $V$ obtained with the device, at $T = 70$ mK. This spectrum has two striking features: first, the very low sub-gap conductance ($G_0 R_N \approx 1/500$), which indicates all bands crossing the Fermi energy are fully gapped. Second, the intricate structure of the quasiparticle peak differs from a standard BCS density-of-states (DOS) by having a relatively low peak and a shoulder at lower energies. The latter feature can be clearly seen in the second derivative (Fig. 1c) where the slope separates into a double peak feature, similar to STS scans of bulk NbSe$_2$[11,20]. Based on this, the flake can be considered bulk in terms of the zero field superconducting properties, and is hence referred to as the bulk sample.

**Two-band model**. Density functional theory calculations[21], and ARPES data[22] show that five independent electronic bands cross the Fermi energy in NbSe$_2$. Of these, four are Nb-derived bands with roughly cylindrical Fermi surfaces, centered at the $\Gamma$ and $K$ points. The fifth is derived from the Se $p_z$ orbitals, which give rise to a small ellipsoid pocket around the $\Gamma$ point. Ref.[11] uses a two-band model to fit NbSe$_2$ tunneling data, which can be justified noting that the Se and Nb-derived bands differ in the density of states and value of the electron-phonon coupling parameter[23]. We follow ref.[11] in fitting our data using the same two-band model, which was developed in various forms by Suhl[24], Schopohl[25], and McMillan[26] (below "SSM").

The model entails a self-consistent solution to the coupled equations for the energy dependent order parameters $\Delta_i(E)$ in the two bands $i$:

$$\Delta_i(E) = \frac{\Delta_i^0 + \Gamma_{ij} \Delta_j(E) / \sqrt{\Delta_j^2(E) - E^2}}{1 + \Gamma_{ij} / \sqrt{\Delta_j^2(E) - E^2} + \Gamma_i^{AG} / \sqrt{\Delta_i^2(E) - E^2}} \quad (1)$$

$\Delta_i^0$ describes the intrinsic gap within each band $i$, that is generated by the electron-phonon coupling and by the scattering rates of quasiparticles between the bands $\Gamma_{ij}$. The extension of the two-band model to include Abrikosov-Gor'kov depairing[27–30]—via the terms with $\Gamma_i^{AG}$—was done by Kaiser and Zuckermann[31]. Here, depairing is due to magnetic field; thus, $\Gamma_i^{AG}$ are set to 0 when no magnetic field is applied. The DOS of each band is then given by

$$N_i(E) = N_i(E_F) \frac{1}{2\pi} \int d\theta \Re \frac{|E|}{\sqrt{(1 + \alpha \cos\theta) \Delta_i^2(E) - E^2}}, \quad (2)$$

where $N_i(E_F)$ is the DOS at the Fermi energy in the normal state in band $i$. The parameter $\alpha = 0.1$, incorporates band-anisotropy. This anisotropy also affects the effective fit temperature ($T = 0.44$ K), which is higher than the sample temperature. Our fit indicates the presence of two independent order parameters. The larger, $\Delta_1^0 = 1.26 \pm 0.01$ meV can be determined with high fidelity. The smaller order parameter $\Delta_2^0$, cannot be determined unambiguously, and could have any value between 0 and 0.3 meV. The suppressed intrinsic superconductivity in the second band is consistent with a band with small density of states and weak electron–phonon coupling. The Se-band, having these properties[23], is thus a candidate for the 2nd band.

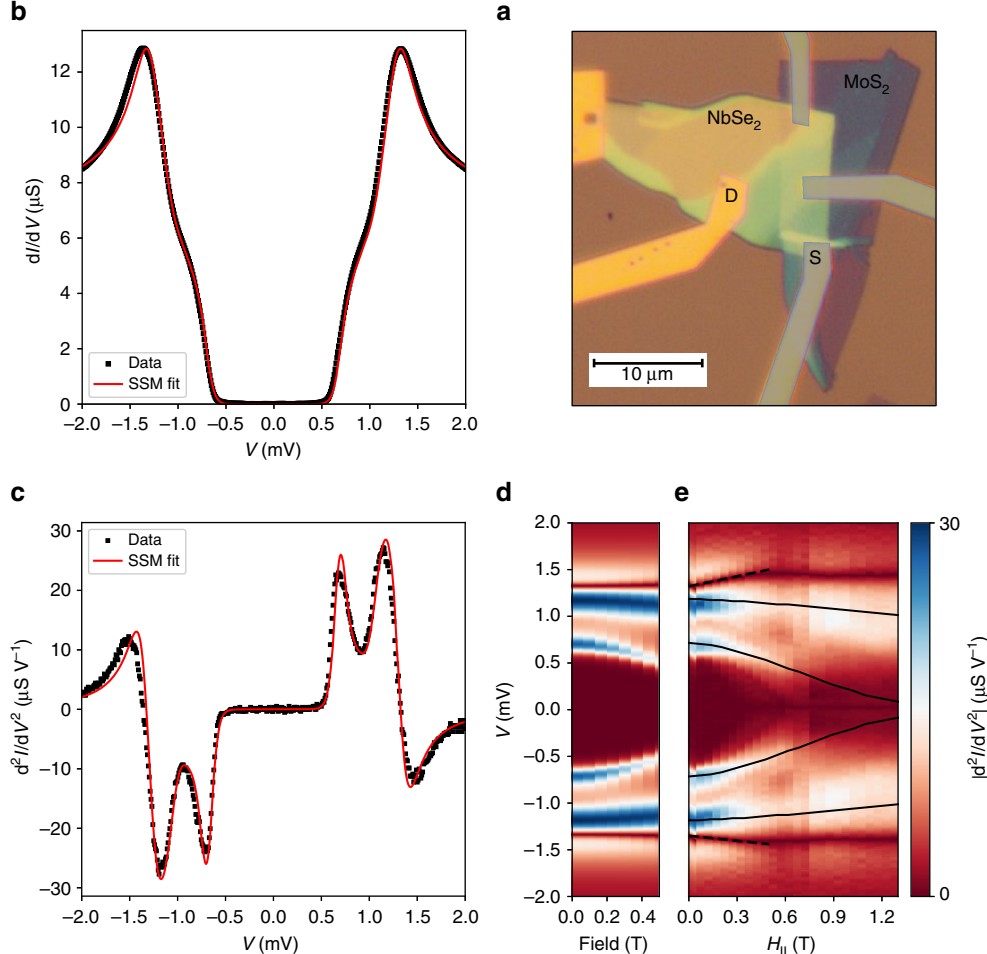

**Fig. 1** Differential conductance of a bulk NbSe$_2$ tunnel device. **a** Optical image of the tunnel junction device. The yellow-green flake is a 50–20 nm thick NbSe$_2$ (20 nm at the source electrode) and the purple-blue flake is a 4–5 layer MoS$_2$. Au electrodes are deposited on the left to serve as ohmic contacts (yellow) and on the right to serve as tunnel electrodes (purple). **b** d$I$/d$V$ vs. $V$ as measured on the device shown in **a** (black) and a fit to the SSM model (red, see details in the text, all fit parameters are given in Supplementary Note 3 and Supplementary Table 1). **c** d$^2I$/d$V^2$ of the data in **b**, and the fit to the SSM model. **d** Simulated $|$d$^2I$/d$V^2|$ as a function of bias voltage and in-plane magnetic field, in the modified SSM model including a field dependent depairing parameter $\Gamma_1^{AG} = 0.05\left(\frac{H_\parallel}{1T}\right)^2$ meV, $\Gamma_2^{AG} = 0.64\left(\frac{H_\parallel}{1T}\right)^2$ meV **e**, measured $|$d$^2I$/d$V^2|$ as a function of in-plane field and voltage bias. The black solid lines follow the positions of the peaks as calculated from the model shown in **d**. The black dashed lines track position of the quasiparticle peak in field, given by d$^2I$/d$V^2 = 0$. The average slope of these lines is 0.27 meV T$^{-1}$

**NbSe$_2$ in in-plane magnetic field: diffusive vs. ballistic regimes.**
We next investigate the evolution of the tunneling spectra vs. in-plane magnetic field $H_\parallel$ (model—Fig. 1d, data—Fig. 1e). The bulk sample appearing in Fig. 1 is thick enough to accommodate Meissner currents leading to orbital depairing. We track the peaks in d$^2I$/d$V^2$, which are associated with the two gaps. The high energy peak in d$^2I$/d$V^2$ depends weakly on the field, whereas the low energy peak evolves nonlinearly towards lower energies. This trend persists up to $H = 0.5$ T which we interpret as $H_{c1}$. Bulk NbSe$_2$ is regarded as a clean-limit superconductor (coherence length $\xi < l$ mean free path). This condition, however, might not hold equally for both bands, since the smaller gap is associated with a larger $\xi$. Indeed, the evolution of the low energy gap in $H_\parallel$ does not agree with the relevant, Doppler shift model[32–34]. Instead, the magnetic field evolution of its d$^2I$/d$V^2$ features towards lower energies is well-reproduced by the diffusive Kaiser-Zuckermann (KZ) model (Fig. 1d), assuming a depairing parameter $\Gamma^{AG}$ quadratic in $H_\parallel$. For a thin sample of thickness $d \ll \lambda$, $\Gamma_i^{AG} = D_i e^2 H_\parallel^2 d^2 / 6\hbar c^2$[35], with $D_i$ the diffusion coefficient, $d \approx 20$ nm and the penetration length $\lambda \approx 230$ nm[36]. We find $\Gamma_2^{AG} = 640$ $\mu$eV at 1 T, corresponding to $D_2 = 40$ cm$^2$ s$^{-1}$. The model

therefore yields a very large value for the diffusion coefficient $D_2 = v_F l/3$, indicating a large Fermi velocity. This lends further support to the identification of this feature with the Se-derived band, where $v_F$ is 4–5 times larger than in the Nb-derived bands[23]. In contrast, the high energy quasiparticle peak moves only slightly and linearly to higher energies at low $H_\parallel$ (dashed line in Fig. 1e). From this, we estimate $v_F \approx 5 \times 10^4$ m s$^{-1}$. This indicates that a comprehensive description of the field evolution of the full spectrum of bulk NbSe$_2$ would require a model with arbitrary disorder bridging clean and diffusive limits.

**Reduced gap in ultra-thin NbSe$_2$.** Our methods allow us to carry out a straightforward comparison of the tunneling spectra from devices with different NbSe$_2$ thicknesses. Figure 2a shows differential conductance curves taken by tunneling into ultra-thin NbSe$_2$ flakes (3 layers, 4 layers), in comparison to the d$I$/d$V$ of the bulk flake discussed above. The spectra of the thin devices are in good agreement with the SSM model (fits are shown in Supplementary Fig. 3, fit parameters are given in Supplementary Table 1). We extract the values of $\Delta_1^0$ from these fits, and

## a

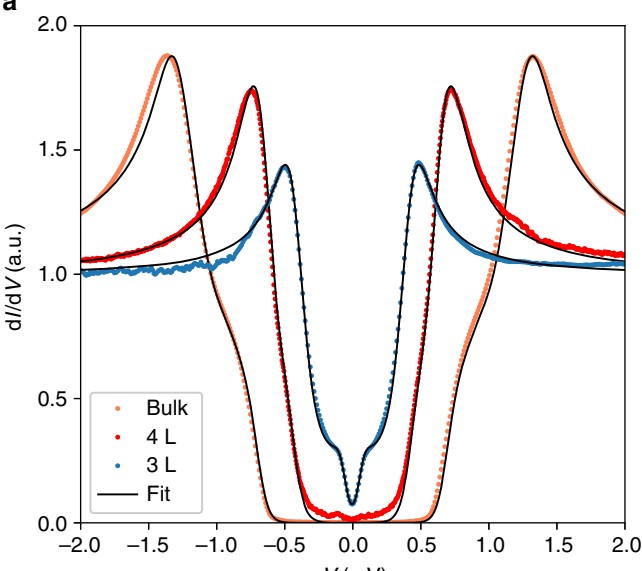

## b

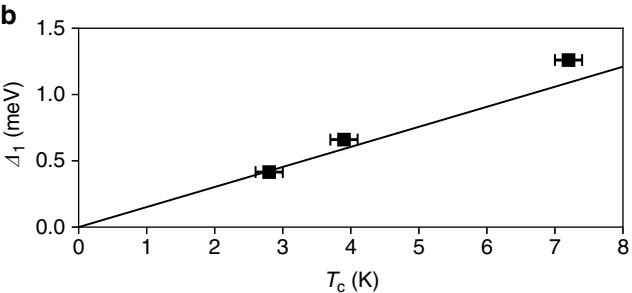

**Fig. 2** Thickness dependent tunneling spectrum. **a** Normalized differential conductance of vdW tunnel junctions with NbSe$_2$ of varying thickness: 20 nm thick (bulk, orange), 4 layers (red) and 3 layers (blue). Each curve is fit to the SSM model (fit parameters reported in Supplementary Note 3).**b** $\Delta_1^0$ of the devices in **a**, extracted from the SSM model, as a function of $T_c$. Solid line shows a comparison to the BCS prediction: $\Delta = 1.76 K_B T_c$. $T_c$ is defined by a 5% reduction of the zero bias conductance relative to the normal conductance of the junctions. The error bars result from the measured hysteresis

separately evaluate $T_c$ using the temperature dependence of the tunneling conductance (details in Supplementary Fig. 4). We find that $\Delta_1^0$ increases with $T_c$ (Fig. 2b), however, deviating slightly from the BCS result $\Delta = 1.76 k_B T_c$. This deviation is plausible due to the multi-band nature of superconductivity in NbSe$_2$. The $T_c$ values measured here are lower than those reported elsewhere[1,37], likely due to higher disorder in our sample[38–40]. Note that, based on ref.[37], it would seem that the $T_c$ dependence on number of layers seen by all works to date is not due to strain or other substrate effects.

**Ultra-thin NbSe$_2$ in magnetic field**. We now turn to the response of the ultrathin devices to in-plane magnetic fields. Figure 2 shows the tunneling spectra for the bulk device (Fig. 2a) and for the 4-layer and 3-layer devices (Fig. 2b and c, respectively). The KZ model, for all 3 devices, is shown in Fig. 2d–f, respectively. In the thin devices, unlike the bulk device, we find that the spectrum changes very little up to 3.5 T (which is the maximal field where a compensation coil keeps a zero perpendicular field). In both 3- and 4-layer devices, there is a small reduction in the height of the

quasiparticle peak; in the 3-layer device, the low energy spectrum exhibits a more intricate evolution (discussed below). Thinner samples have shorter scattering lengths and longer coherence lengths, and should thus be closer to the diffusive limit. Indeed, the KZ model fits our data well, and we use it to quantify the reduction of the peak height of the 3- and 4-layer devices (Fig. 2e, f), finding that the depairing term $\Gamma_1^{AG} \approx 0.5\,\mu eV$ at 1 T (all model parameters are given in Supplementary Table 2).

Since orbital depairing is quadratic in sample thickness, we expect it to be diminished in the 3- and 4-layer devices, allowing us to probe the spin-dependent interaction. The interaction of the spin with magnetic field should lead to Zeeman splitting of the spectrum, which we do not see. This could be due to two mechanisms. First, spin–orbit scattering can effectively randomize the spin, giving rise to a depairing parameter given by $\Gamma^{AG} = \tau_{SO}e^2\hbar H_\parallel^2/2m^2$,where $\tau_{SO}$ is the spin orbit scattering time[35]. Second, ISOC can align the spins in the out-of-plane direction with an effective field, $H_{SO}$, and the depairing term $\Gamma^{AG} \sim 2\mu_B H_\parallel^2/H_{SO}$[1,5]. The first scenario can be ruled out, since it yields $\tau_{SO} < 50$ fs, shorter than the scattering time[5]. The ISOC case is more likely, and the depairing energy of 0.5 µeV at 1 T (for the 3L device) yields $H_{SO} \approx 200$ T. Using $H_P = 4.9\,T$ (extracted by setting $\Delta_1 = 0.4$ meV) we can estimate $H_{c2} = \sqrt{H_{SO}H_P} \approx 30$ T, consistent with transport experiments[1]. We note that since the orbital term is not entirely suppressed, and we cannot estimate its contribution to the depairing, the estimate for $H_{SO}$ is a lower bound. Further details concerning the possible interpretation of the depairing term are given in Supplementary Note 3 and Supplementary Table 2.

The stability of the larger gap can even be demonstrated above the Pauli limit, $H_P = 4.9\,T$ for the 3-layer device. In Fig. 3c we present the density of states of this sample above the Pauli limit by applying an in-plane field of 6.4 T. At this field, our measurement system did not allow us to compensate for angle misalignment leading to small component of perpendicular field ($\approx 0.2$ T) and possible vortex penetration. Nevertheless, the size of the gap remains unchanged. This lends further support to ISOC as the mechanism protecting NbSe$_2$ superconductivity at high parallel fields.

**Anomalous secondary gap**. The sub-gap spectrum of the 3-layer device appears to exhibit a secondary, well-formed small gap of $\Delta_2 = 50\,\mu V$, which is suppressed at $H_\parallel = 1.2$ T. As we show in Fig. 3c, f, the KZ model reproduces this data remarkably well. Here, too, the depairing term is quadratic in $H_\parallel$, with $\Gamma_2^{AG} \approx 13\,\mu eV$ at 1 T. This value is too big to be interpreted in terms of orbital depairing. The observed depairing could alternatively be associated with spin-orbit-driven spin-flip scattering, with $\tau_{SO} \approx 1.5$ ps. This depairing shows that unlike the band with larger gap, this band is not protected by ISOC. This difference can be explained by associating the smaller order parameter with the Se-derived band. The outer gap, appearing immune to depairing, would then be associated with the Nb-derived $K$-band. We note that this interpretation leaves open the question of the role of the Nb-derived $\Gamma$-band. Addressing this question will require further spectral studies—in particular, of monolayer NbSe$_2$ where the Se-derived band does not cross the Fermi energy.

**Discussion**
Our results show that TMD semiconductors transferred on top of NbSe$_2$ form stable tunnel barriers with a hard gap. We show that the SSM model, modified to include diffusive depairing, successfully reproduces the tunneling spectra in both the bulk and the ultrathin limits, in the presence of in-plane magnetic fields. This allows us to probe the effect of the spin and orbital degrees

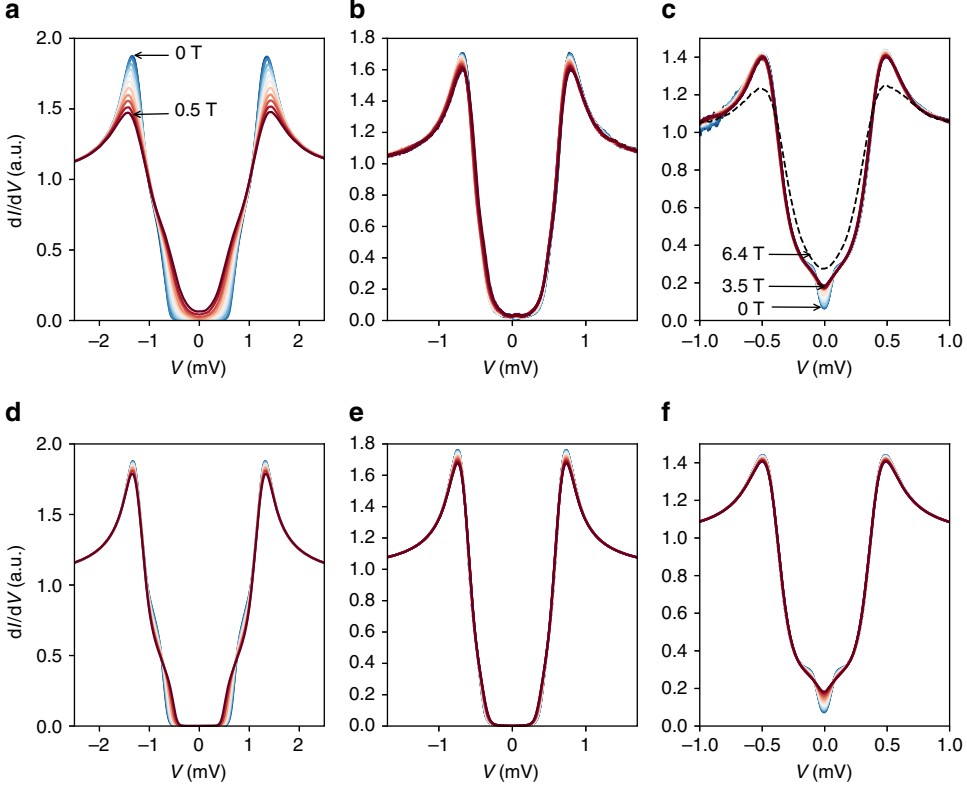

**Fig. 3** Response of the tunneling conductance to parallel magnetic fields. **a** d$I$/d$V$ curves at increasing magnetic field parallel to the NbSe$_2$ layers ($H_∥$) of the bulk sample, at $0 < H_∥ < 0.5$ T. **b** Same for the 4-layer device at $0 < H_∥ < 3.5$ T, and **c** same for the 3-layer device. Dashed line in **c**: d$I$/d$V$ curve taken with the 3-layer device at uncompensated parallel field of 6.4 T. **d**–**f** d$I$/d$V$ calculated from the KZ model for the bulk, 4-layer and 3-layer, respectively. Model parameters are given in Supplementary Note 3

of freedom on the spectra, thereby differentiating between the responses of the different bands to the field. The large gaps, in the 3- and 4-layer devices, are remarkably stable to depairing by the in-plane field, exhibiting very small depairing energies (<1 µeV), which place a tight cap on the spin-depairing observed on this band, lending support to ISOC as the mechanism behind this stability. We suggest that our technique can be generalized to work with many other material systems, such as organic (super) conductors and other fragile systems which have hitherto not been investigated using tunneling spectroscopy.

## Methods

**Device fabrication and measurement**. The vdW tunnel junctions were fabricated using the dry transfer technique[41], carried out in a glove-box (nitrogen atmosphere). NbSe$_2$ flakes were cleaved using the scotch tape method, exfoliated on commercially available Gelfilm from Gelpack, and subsequently transferred to a SiO$_2$ substrate. MoS$_2$ and WSe$_2$ flakes were similarly exfoliated and thin flakes suitable for the formation of tunnel barriers were selected based on optical transparency. The barrier flake was then transferred and positioned on top of the NbSe$_2$ flake at room temperature. Ti/Au contacts and tunnel electrodes were fabricated using standard e-beam techniques. Prior to the evaporation of the ohmic contacts the sample was ion milled for 15 s. No such treatment was done with the evaporation of the tunnel electrodes. All transport measurements were done in a $^3$He–$^4$He dilution refrigerator with a base temperature of 70 mK. The AC excitation voltage was modulated at 17 Hz; its amplitude was 15 µV at all temperatures for the bulk device and 10 µV for the few-layer devices. Measurement circuit details are provided in Supplementary Fig. 5.

**Data availability**. The data that support the findings of this study are available from the corresponding author upon request.

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

## Acknowledgements

We thank P. Février and J. Gabelli for helpful discussions on tunnel barriers, and T. Cren for the same on $NbSe_2$. This work was funded by a Maimondes-Israel grant from the Israeli-French High Council for Scientific & Technological Research and by an ANR JCJC grant (SPINOES) from the French Agence Nationale de Recherche. H.S. acknowledges support by ERC-2014-STG Grant No. 637298 (TUNNEL) and Marie Curie CIG Grant No. PCIG12-GA-2012-333620. T.D. is grateful to the Azrieli Foundation for an Azrieli Fellowship. F.M. has received funding from the European Union's Horizon 2020 research and innovation programme under the Marie Skodowska-Curie Grant No. 659247. L.A. and M.K. are supported by the Israeli Science Foundation through Grant No. 1287/15.

## Author contributions

T.D. fabricated the devices. C.Q.H.L., T.D., and M.A. performed the measurements. T.D., F.M, L.A., M.K., C.Q.H.L., M.A., and H.S. contributed to data analysis and the writing of the manuscript.

## Additional information

**Competing interests:** The authors declare no competing financial interests.

