## [Peer Review File · Nature Communications]

Reviewers' comments:

Reviewer #1 (Remarks to the Author):

Comments on the manuscript: "Hard superconducting gap and vortex-state spectroscopy in NbSe₂ van der Waals tunnel junctions", by T. Dvir et al.

The authors report a method to build planar tunneling junctions based on the use of Van der Waals semiconductors as the insulating layer. This method is within the current state of the art and has been used for several years already in other compounds. They provide a proof of the concept in superconducting junctions by performing tunneling measurements in 2H-NbSe₂, a well-known s-wave BCS superconductor, and claim that the method can be extended to many other superconducting materials.

Although the method provides nice results, this material has been extensively measured using tunneling measurements, including planar junctions. I do not appreciate any novel or advanced insight in the manuscript. All results presented by the authors are already known by the community. Therefore, I do not believe that the manuscript meets the required criteria for publication and broad readership in Nature Communications. Instead, I would suggest the authors to submit a modified manuscript to a more technical journal elsewhere.

Below I provide a more detailed justification to support my opinion:

- The authors claim that the fit shown in Fig. 1 "is remarkably precise". However they use as fitting parameter a temperature of 0.52 K which is almost an order of magnitude higher than the experimental temperature of 70 mK. They speculate that such a large temperature difference is due to the presence of a gap anisotropy in the reciprocal space. This is equivalent to consider a distribution of gaps. Such an approach has been already used to fit tunneling spectra in NbSe₂ at millikelvin temperatures, also considering intrinsic superconducting pairing in both bands (see for instance, Physica C 404 306-310 2004 and or PRB 77 134505 2008). Not only tunneling experiments, but also further important experimental evidence for intrinsic two-band superconductivity with two well defined values for the superconducting gaps has been reported using other techniques, whose references are completely missing in the manuscript. For instance, thermodynamic measurements (heat capacity, thermal conductivity; see PRB 76 212504 2007 and PRL 90 117003 2003), London penetration depth (PRL 98 057003 2007), dHvA (Phys. Cond. Matt. 6 4479 1994) and even ARPES (Nature Physics 3 720 2007) which provide direct measurement of a gap opening in different bands. Thus, I do not find any new insight here.

- Vortex bound states have been largely reported previously in STM measurements with real space resolution. I really cannot understand what new insight can be obtained from planar junction measurements, which provide spatially averaged information. This point is not addressed in the manuscript that instead speculates with a magnetic field dependence of several parameters that do not provide any conclusive insight.

- Besides, the authors do not mention the strong sixfold-in-plane anisotropy of the vortex core states. This was first reported by Hess et al in pioneering tunneling experiments in NbSe₂ [PRL 64 2711 1990] and by many others later. As a result of these studies it is known that vortex cores have a star shape with a higher bound state spectral weight and a vortex overlap along high symmetry directions. The effect of the in-plane anisotropy of vortex core states on the tunneling data taken under magnetic field has not been taken into account in the manuscript. This effect would be particularly important on the data taken in presence of an in-plane magnetic field. The measured spectrum should depend on the in-plane direction of the magnetic field with respect to the in-plane crystalline axis. The authors seem to apply the in-plane field along a random direction or, at least, this issue is not specified in the manuscript. A study taking into account the in-plane

angular dependence of the superconducting density of states will be required to provide useful insight in this material.

- In the supplementary information S7, the sub-gap conductance at zero field is discussed. An SSM model + gamma Dynes and a BTK (with a single gap!) are considered. Why using a single Delta for BTK if they have shown that two gaps (bands) are involved? Finally, the effect is suggested to be due to the environment, but no serious analysis is done to support this suggestion. Instead of this, a trivial comparison with the behaviour of Al/Al₂O₃/Al NIS(????) junctions measured in a similar environment is reported.

- Can the authors discard the possibility of gold pinholes as the origin of a parallel resistance to the junction?

- In fig.2d are shown for comparison the results for three different devices. Device C presents a different behaviour from the two others, and no comment about this discrepancy is referred.

- It is intriguing for this referee that the experimental data reported in Figure S6 show symmetrical structures around zero bias. Have they measured for positive and negative bias, in which case the structures should have a certain physical meaning, or simply they have duplicated the data obtained for a single bias direction?

Reviewer #2 (Remarks to the Author):

The authors investigate a new tunneling junction between a normal metal and superconductor. By using van der Waals semiconductors MoS₂ and WSe₂ they succeeded in forming a good quality NIS junction. This is proved by measuring the tunneling characteristics I(V) of 2H-NbSe₂ under both perpendicular and parallel fields to the junction plane. The obtained I(V) reproduces the known two gap structure in this compound whose temperature and field evolution clearly demonstrate how good junction is formed by this technique. Although in this paper only known tunneling spectra on 2H-NbSe₂ are reproduced, thus there is little new physical information reported here, this novel technique based on van der Waals semiconductors may potentially serve giving rise to new physics in future.

Thus I recommend that this paper is publicshable.

I notice the following minor errors:

- (1) In Fig.2 (d): the label (d) is missing.
- (2) p 7 last line: ...NbSe₂ [49]. Ref. [42] seems to be not relevant here.
- (3) Ref. [24] miss spelling of the author's name.
- (4) Ref. [54] NbSe₂ should be lower case.

Reviewer #3 (Remarks to the Author):

The authors present tunneling spectroscopy data on superconductor NbSe₂ samples based on van der Waals tunnel junctions constructed from thin layers of MoS₂ and WSe₂. The junctions are very high quality allowing precision low-temperature measurements of the NbSe₂ density of states. The authors observe a hard superconducting gap and a well-resolved two-band quasiparticle peak structure with strong evidence of intrinsic superconductivity in both bands. Measurement in an applied magnetic field show linear sub-gap structure consistent with vortex bound states. The data are high quality and the subsequent analyses are well-motivated and clearly explained. More generally, these measurements demonstrate that van der Waals materials can be useful to form high-quality tunneling barriers in systems for which oxide barriers are not readily achieved. I recommend publication in Nature Communications.

Response to reviewers

Reviewer #1

We thank the reviewer for his/her careful reading of the manuscript.

We have carefully considered the reviewer's main criticism concerning the novelty of our work. We believe that the ability to acquire data with such high resolution using ex-situ methods which are applicable to a large variety of materials is indeed novel, albeit from a more technological point of view. Nevertheless, we have further conducted an extended series of measurements and have obtained the first tunnelling spectra of 3- and 4-layer NbSe₂, and the evolution of the superconducting gap vs. critical temperature. Furthermore, we find that the superconducting gap in a thin (3-layer) device is immune to the application of planar magnetic fields beyond the Pauli limit, lending support to the claim that these materials exhibit Ising superconductivity.

The ultra-thin limit of NbSe₂ (and TMDs in general) has been attracting a lot of attention recently, as superconductivity in these materials is expected to be 'Ising': Ising spin-orbit coupling, which is due to the lack of inversion symmetry in the plane of monolayer NbSe₂, locks spins in the K and K' valleys out of plane and in opposite directions. Cooper pairs formed from electrons in opposite valleys are predicted to survive in in-plane magnetic fields beyond the Pauli (or Clogston-Chandrasekar) limit, and indeed zero-resistance states have been observed very recently in mono- and few-layer NbSe₂. Our data now provide the first spectroscopic evidence for Ising superconductivity.

Please note that, in order to keep the manuscript to a reasonable length, and also to ensure a clear and coherent message, the following points have been moved to Supplementary Information or removed. If the reviewer (or editor) feels they should be put back in, we would be more than happy to do so.

- The discussion of the SSM model has been shortened, and details moved to the Supplementary Information. We now emphasize the main conclusions: the two-band model fits the data very well, and there are two intrinsic superconducting order parameters (as opposed to a single intrinsic superconducting order parameter where the second one is induced).
- The discussion of perpendicular magnetic field has been removed.
- The discussion of the in-plane magnetic field has changed its focus. Previously we discussed the H_{c1} phase transition, in which vortices begin to penetrate the sample. In the revised version, we compare the response of devices of different thicknesses.

Detailed responses to specific points raised by the reviewers follow.

Comments on the manuscript: "Hard superconducting gap and vortex-state spectroscopy in NbSe₂ van der Waals tunnel junctions", by T. Dvir et al.

The authors report a method to build planar tunneling junctions based on the use of Van der Waals semiconductors as the insulating layer. This method is within the current state of the art and has been used for several years already in other compounds. They provide a proof of the concept in superconducting junctions by performing tunneling measurements in 2H-NbSe₂, a well-known s-wave BCS superconductor, and claim that the method can be extended to many other superconducting materials.

Although the method provides nice results, this material has been extensively measured using tunneling measurements, including planar junctions. I do not appreciate any novel or advanced insight in the manuscript. All results presented by the authors are already known by the community. Therefore, I do not believe that the manuscript meets the required criteria for publication and broad readership in Nature Communications. Instead, I would suggest the authors to submit a modified manuscript to a more technical journal elsewhere.

The referee argues that tunnel barriers based on vdW materials is within the state-of-the art, and have been used for tunnelling into other (non-superconducting) compounds. In fact, we are only aware of tunnelling devices realized on graphene. A Josephson junction, where both counter-electrodes are metals, was studied in Island et al. We are not aware of tunnelling experiments, using vdW barriers, carried out on superconductors.

Second, the referee claims that NbSe₂ was extensively studied by tunnelling, “including planar junctions”. Indeed, NbSe₂ was extensively studied by STM. Device-based planar junctions on NbSe₂ have only been reported once, to our knowledge, by Morris and Coleman in Physics Letters 43, 11 (1973). In this work, the barrier appears to be of poor quality, i.e. highly transparent. We are not aware of any work on NbSe₂ where a high quality, planar oxide-based junction is investigated.

Below I provide a more detailed justification to support my opinion:

- The authors claim that the fit shown in Fig. 1 “is remarkably precise”. However they use as fitting parameter a temperature of 0.52 K which is almost an order of magnitude higher than the experimental temperature of 70 mK. They speculate that such a large temperature difference is due to the presence of a gap anisotropy in the reciprocal space. This is equivalent to consider a distribution of gaps. Such an approach has been already used to fit tunneling spectra in NbSe₂ at millikelvin temperatures, also considering intrinsic superconducting pairing in both bands (see for instance, Physica C 404 306-310 2004 and or PRB 77 134505 2008).

The referee claims that our approach is equivalent to the use of a distribution of gaps. The two approaches – a distribution of gaps and two bands – are in fact *inequivalent*. In the first approach, used in the Physica C paper among others, a BCS DOS is assumed, and the entire shape of the density of states is claimed to arise from the inhomogeneity/anisotropy of the gap, whether in real or momentum space. In the 2nd approach, which we apply here, the main shape of the density of states is assumed to be given by the presence of two gaps (described by the SSM model). Within this two-gap model, a slight peak broadening can arise due to several sources, all of which are discussed in detail in the manuscript.

We nevertheless emphasise that the overall shape of the DOS we measure points quite clearly to the presence of two gaps rather than a distribution. We do not claim the latter and hope that the revised manuscript makes this point more clearly.

Not only tunneling experiments, but also further important experimental evidence for intrinsic two-band superconductivity with two well defined values for the superconducting gaps has been reported using other techniques, whose references are completely missing in the manuscript. For instance, thermodynamic measurements (heat capacity, thermal conductivity; see PRB 76 212504 2007 and PRL 90 117003 2003), London penetration depth (PRL 98 057003 2007), dHvA (Phys. Cond. Matt. 6 4479 1994) and even ARPES (Nature Physics 3 720 2007) which provide direct measurement of a gap opening in different bands. Thus, I do not find any new insight here.

Indeed, whether the superconducting state in NbSe₂ has one or two gaps is a long-standing controversy with quite a lot of literature on both sides. We thank the reviewer for drawing our attention to ref PRB. 77 134505 (2008) which has measured the tunnelling density of states of NbSe₂ with the highest resolution we are aware of and clearly shows two features visible in the second derivative, as does our data. Both lend weight to the two-gap scenario. The novelty of our work, specifically for the bulk sample, is in the analysis with the SSM model, which points to the contribution of second intrinsic order parameter. This is possible thanks to the high resolution of our data.

We agree with the referee that two-band superconductivity in NbSe₂ has long been investigated. As this is not a central theme in the present version, we refer to the wealth of prior work by pointing to citations in Noat et al. who present a thorough list of recent evaluations of the gap structure.

- Vortex bound states have been largely reported previously in STM measurements with real space resolution. I really cannot understand what new insight can be obtained from planar junction measurements, which provide spatially averaged information. This point is not addressed in the manuscript that instead speculates with a magnetic field dependence of several parameters that do not provide any conclusive insight.

Our ability to observe such vortices is yet another indication to the high quality of the barrier, resulting in a hard gap. We agree with the reviewer that this is a point which is more technological. As we have changed the focus of the manuscript, the spectroscopy of vortex bound states no longer holds a central role in this work.

- Besides, the authors do not mention the strong sixfold-in-plane anisotropy of the vortex core states. This was first reported by Hess et al in pioneering tunneling experiments in NbSe₂ [PRL 64 2711 1990] and by many others later. As a result of these studies it is known that vortex cores have a star shape with a higher bound state spectral weight and a vortex overlap along high symmetry directions. The effect of the in-plane anisotropy of vortex core states on the tunneling data taken under magnetic field has not been taken into account in the manuscript. This effect would be particularly important on the data taken in presence of an in-plane magnetic field. The measured spectrum should depend on the in-plane direction of the magnetic field with respect to the in-plane crystalline axis. The authors seem to apply the in-plane field along a random direction or, at least, this issue is not specified in the manuscript. A study taking into account the in-plane angular dependence of the superconducting density of states will be required to provide useful insight in this material.

As previously mentioned, vortices are no longer a focus of the revised manuscript.

- In the supplementary information S7, the sub-gap conductance at zero field is discussed. An SSM model + gamma Dynes and a BTK (with a single gap!) are considered. Why using a single Delta for BTK if they have shown that two gaps (bands) are involved? Finally, the effect is suggested to be due to the environment, but no serious analysis is done to support this suggestion. Instead of this, a trivial comparison with the behaviour of Al/Al₂O₃/Al NIS(????) junctions measured in a similar environment is reported.

In Supplementary Section 7, we discuss several mechanisms for the very small but non-zero conductance below the gap. Andreev processes can account for that, and this is taken into account in the BTK model. We are unaware of an extension of the BTK model to the case of two gap superconductors, and in particular to the SSM model. In any case, at low sub-gap energies, the SSM model in our parameter range converges with a the BCS model. Thus, taking a single gap BTK at this voltage regime is justified, especially for making an estimate.

Regarding the effects of the environment, we refer the reviewer to a thorough discussion of this matter by Pekola et al. (PRL, 105, 026803 (2010)), whom we cite. We used their results to draw a comparison between other junctions measured at the same environment and our sample. The Al/Al₂O₃/Al in question is indeed an NIS junction as one of the electrodes is 100nm thick and is turned normal by a small in-plane field (less than 100mT) whereas the other, which is 6nm thick remains superconducting up to above 3T. We have made this point clearer in the revised manuscript.

- Can the authors discard the possibility of gold pinholes as the origin of a parallel resistance to the junction?

We cannot exclude the possibility of gold pinholes or any other source of parallel resistance, as is clearly stated in the manuscript. We note again that the very small sub-gap conductance is consistent with environmental noise for the reasons stated in the manuscript.

- In fig.2d are shown for comparison the results for three different devices. Device C presents a different behaviour from the two others, and no comment about this discrepancy is referred.

As vortices are no longer a focus of the revised manuscript, this figure has been removed.

- *It is intriguing for this referee that the experimental data reported in Figure S6 show symmetrical structures around zero bias. Have they measured for positive and negative bias, in which case the structures should have a certain physical meaning, or simply they have duplicated the data obtained for a single bias direction?*

This is an artefact resulting from our data analysis process. To avoid this ambiguity, we have replotted all of the figures with data as taken or divided by a constant factor. We further added the following text to Supplementary 1: “To fit the resulting dI/dV s to the SSM model as discussed in the main text, the data was horizontally shifted to account for zero bias drift, and divided by a dI/dV curve taken at $T > T_c$. It was then symmetrized around zero bias, and fitted using least-squares method to the SSM model. The data shown in the figures, in the main text and in the supplementary, is either the original data as measured, or the data divided by a constant normalization factor.”

Reviewer #2:

The authors investigate a new tunneling junction between a normal metal and superconductor. By using van der Waals semiconductors MoS₂ and WSe₂ they succeeded in forming a good quality NIS junction. This is proved by measuring the tunneling characteristics $I(V)$ of 2H-NbSe₂ under both perpendicular and parallel fields to the junction plane. The obtained $I(V)$ reproduces the known two gap structure in this compound whose temperature and field evolution clearly demonstrate how good junction is formed by this technique. Although in this paper only known tunneling spectra on 2H-NbSe₂ are reproduced, thus there is little new physical information reported here, this novel technique based on van der Waals semiconductors may potentially serve giving rise to new physics in future. Thus I recommend that this paper is publishable.

I notice the following minor errors:

- (1) In Fig.2 (d): the label (d) is missing.*
- (2) p 7 last line: ...NbSe₂ [49]. Ref. [42] seems to be not relevant here.*
- (3) Ref. [24] miss spelling of the author's name.*
- (4) Ref. [54] NbSe₂ should be lower case.*

We thank the reviewer for his/her remarks and are glad that s/he found the original manuscript publishable. Indeed, in original manuscript we focused on the merits of the tunnel junction. We have now made substantial revisions, which include observations of new physics in few-layer NbSe₂. We believe that these should increase the relevance and impact of the manuscript.

Reviewer #3:

The authors present tunneling spectroscopy data on superconductor NbSe₂ samples based on van der Waals tunnel junctions constructed from thin layers of MoS₂ and WSe₂. The junctions are very high quality allowing precision low-temperature measurements of the NbSe₂ density of states. The authors observe a hard superconducting gap and a well-resolved two-band quasiparticle peak structure with strong evidence of intrinsic superconductivity in both bands. Measurement in an applied magnetic field show linear sub-gap structure consistent with vortex bound states. The data are high quality and the subsequent analyses are well-motivated and clearly explained. More generally, these measurements demonstrate that van der Waals materials can be useful to form high-quality tunneling barriers in systems for which oxide barriers are not readily achieved. I recommend publication in Nature Communications.

We thank the reviewer for his/her kind remarks.

Reviewers' comments:

Reviewer #1 (Remarks to the Author):

Authors have significantly changed their paper. They have removed the results on the vortex lattice and given less weight to their discussion on the two-gap properties in NbSe₂. Instead, they address a totally new topic on Ising superconductivity that was not even mentioned in the previous version.

I do agree with the other referees' reports that the technical advances made by the authors in the development of planar junctions with tunneling barriers showing very high transparencies are significant and would allow them to provide new insight into interesting topical problems in layered materials.

However, this is not the case in neither of the two versions submitted by the authors. In the present version authors report enhanced superconductivity in few layer NbSe₂ under magnetic field parallel to the layers. This result was already reported in Ref. [1] of the manuscript not only in few layer NbSe₂ but also in monolayer NbSe₂ using transport measurements in a much wider range of magnetic fields exceeding H_P over 6 times. Here the authors use a different technique, tunneling measurements, to essentially find the same result providing one single curve slightly above H_P . There is not new insight but confirmation of previous result using other technique. Moreover, as authors mention in the introduction, it is not understood how Ising superconductivity can appear when inversion symmetry is recovered in few layer compounds. The authors do not discuss this issue nor provide any new understanding at all.

Therefore, I cannot recommend the present version this paper for publication in Nature Communications.

Finally, I suggest the authors to take into account the following comments before publication:

- Authors estimate the value for the superconducting gap, Δ , from the energy position of quasiparticles peaks. Δ is better approached by the inflection point of the density of states given by the maxima/minima in the derivative of the density of states. Using this value for the superconducting gap the authors will likely find that the ratio Δ/k_{BT_c} is closer to the BCS value instead of speculating with other sources for the deviation of this ratio such as strong electron-phonon coupling.

- Authors show evolution of a single gap with the sample thickness in Fig. 2b. It would be much more useful if, instead, the authors show the evolution of the two superconducting gaps they found in bulk samples. Do the two superconducting gaps decrease with the magnetic field in a similar way? How does the two gap structure evolve with in-plane magnetic field? What is its role in the Ising superconductivity? This would indeed provide new insight to the community.

- The latter maybe related or not to the appearance of a small gap in the density of states of the three layer sample. It would be desirable that authors explain the origin of such a feature. Have they reproduced these results in different junctions?

I sincerely believe that authors have in hand a tool to give a physical answer to these and other relevant questions.

Response to Reviewer 1, Manuscript NCOMMS-17-11423-T, Dvir et al.

We present a major revision with respect to the previous version, with an addition of a two-band depairing model which is used for fitting the in-plane magnetic field data. This allows us to compare to existing transport results, and provide extra insight regarding the role of the smaller band in NbSe₂ superconductivity. This was done following comments made by Referee 1, whom we thank for taking the time to read our manuscript and suggest this direction.

Specifically, we have revisited the in-plane tunneling datasets for the bulk and thin devices. To analyze these data, we have modified the two-band model, used in earlier versions of the manuscript, to include the effect of depairing on each band. This analysis yields a quantitative evaluation of the depairing parameters for the thin and thick samples. In the thin samples we find a strongly suppressed depairing, which is the combined effect of the spin and orbital degrees of freedom. As a result, our data can now shed light on the role of the different bands in the depairing process.

The main changes to the manuscript are the following:

Figure 1: We have added a panel reporting d^2I/dV^2 vs. in-plane magnetic field. Together with the data, we plot simulated d^2I/dV^2 based on an orbital depairing model.

Figure 2: We have changed the way the gap Δ_1 is calculated, now using the two-band model

Figure 3: Upgraded to include simulated plots, based on the depairing model.

Point-by-point response

Authors have significantly changed their paper. They have removed the results on the vortex lattice and given less weight to their discussion on the two-gap properties in NbSe₂. Instead, they address a totally new topic on Ising superconductivity that was not even mentioned in the previous version.

I do agree with the other referees' reports that the technical advances made by the authors in the development of planar junctions with tunneling barriers showing very high transparencies are significant and would allow them to provide new insight into interesting topical problems in layered materials.

Note: We show barrier of very low transparency, allowing for the detailed measurement of the sub-gap spectrum.

However, this is not the case in neither of the two versions submitted by the authors. In the present version authors report enhanced superconductivity in few layer NbSe₂ under magnetic field parallel to the layers. This result was already reported in Ref. [1] of the manuscript not only in few layer NbSe₂ but also in monolayer NbSe₂ using transport measurements in a much wider range of magnetic fields exceeding H_P over 6 times. Here the authors use a different technique, tunneling measurements, to essentially find the same result providing one single curve slightly above H_P . There is not new insight but confirmation of previous result using other technique.

We believe that there are several new insights present in this modified version of the manuscript:

1. We show that van der Waals insulators can serve as high quality tunnel barriers for superconductors. We demonstrate that a hard gap can be obtained, and that it is possible to probe the density of states at a resolution of a few μV . We are unaware of other *ex situ* techniques that allow measurement at this resolution.
2. We measure for the first time the density of states of ultrathin NbSe₂ allowing for a direct measurement of the energy gap. In two-band superconductors the energy gap and T_c are not simply related. We show that the ratio of the larger order parameter and T_c is not precisely given by the BCS ratio.
3. We show that the two band nature of NbSe₂ persists at the thin limit, by fitting the data to a theoretical two band model.
4. We show that the two bands in bulk NbSe₂ respond differently to the application of in-plane magnetic field and argue that this is the result of a difference in the Fermi velocity.
5. We show the effect of depairing in ultrathin NbSe₂. This indeed gives further validity for the picture of Ising superconductivity. We note that unlike transport measurements, which are based on a single data point at high magnetic field ($B_{c2}(T)$), we see this effect over a range of magnetic fields.

Moreover, as authors mention in the introduction, it is not understood how Ising superconductivity can appear when inversion symmetry is recovered in few layer compounds. The authors do not discuss this issue nor provide any new understanding at all.

Our tunneling data shows that even at the 3-4 layer limit, very limited depairing is present. This points to a significant role of the Ising mechanism even at the few layer limit. Furthermore, we can associate different depairing processes with the different gaps. The smaller gap, for example, is identified with the Se-derived band. This, in turn, suggests that the protected bands are both the K- and Γ -centered Nb-derived bands.

While this does not directly address the question of broken/restored inversion symmetry, it does provide further insight into the problem.

Finally, I suggest the authors to take into account the following comments before publication:

- Authors estimate the value for the superconducting gap, Delta, from the energy position of quasiparticles peaks. Delta is better approached by the inflection point of the density of states given by the maxima/minima in the derivative of the density of states. Using this value for the superconducting gap the authors will likely find that the ratio Δ/k_{BT_c} is closer to the BCS value instead of speculating with other sources for the deviation of this ratio such as strong electron-phonon coupling.

We thank the referee for pointing this out. Following this comment, we re-estimated the SC gap, Delta, by fitting the spectrum with the two-band model. We believe that this is the most accurate way to measure this value. Figure 2 was revised accordingly, and this is discussed in the relevant section of the revised manuscript.

- Authors show evolution of a single gap with the sample thickness in Fig. 2b. It would be much more useful if, instead, the authors show the evolution of the two superconducting gaps they found in bulk samples.

We fit all three samples to the two-band model, however, while the order parameter of the larger band can be determined unambiguously, this is not so with the smaller order parameter, which can be determined as a sum with one of the scattering ratios. We show that the two band is applicable also at the ultra-thin limit. This is mentioned in the main text, and the details of the fitting are given in supplementary section 3.

Do the two superconducting gaps decrease with the magnetic field in a similar way? How does the two gap structure evolve with in-plane magnetic field?

In the bulk sample we find the two gaps respond **differently** to field as a result of different Fermi velocity. This effect is likely a consequence of a far greater Fermi velocity associated with the smaller gap. As we explain in the main text, the smaller and larger gap are markedly different, not only in their dynamics, but also in the type of depairing they experience. Specifically, the large gap is likely in the clean limit, and exhibits a Doppler shift in energy. The smaller gap agrees well with the diffusive description. Such coexistence of clean and diffusive limits in the multi-gap NbSe₂ system points, in our opinion, to interesting future questions related to this material.

What is its role in the Ising superconductivity? This would indeed provide new insight to the community.

By using the modified two-band model, we show that the depairing observed in the data is minimal. It cannot be associated with spin-randomization by spin-orbit scattering, and is likely shared by the orbital term and the spin processes.

In the modified version on the manuscript, these points are discussed in detail.

- The latter maybe related or not to the appearance of a small gap in the density of states of the three layer sample. It would be desirable that authors explain the origin of such a feature. Have they reproduced these results in different junctions?

We have so far been able to fabricate only a single 3-layer device. The small energy gap observed in this device does not appear in the thicker devices. However, it serves as an example of a gap that undergoes depairing with field which originates from the spin degree of freedom, as we show in Figure 3(c,f). This suggests that this gap is associated with a band which is not protected by Ising spin orbit coupling, perhaps with the Se-derived band. This is discussed in the final part of the revised manuscript.

I sincerely believe that authors have in hand a tool to give a physical answer to these and other relevant questions.